# Undernutrition, food insecurity, and leprosy in North Gondar Zone, Ethiopia: A case-control study to identify infection risk factors associated with poverty

Puneet Anantharam[1], Lisa E. Emerson[1], Kassahun D. Bilcha[2], Jessica K. Fairley[1,3]*, Annisa B. Tesfaye[4]

1 Hubert Department of Global Health, Rollins School of Public Health, Emory University, Atlanta, Georgia, United States of America, 2 Department of Dermatology, Emory University School of Medicine, Atlanta, Georgia, United States of America, 3 Department of Medicine, Division of Infectious Diseases, Emory University School of Medicine, Atlanta, Georgia, United States of America, 4 University of Gondar, College of Medicine and Health Sciences, Gondar, Ethiopia

* jessica.fairley@emory.edu

## Abstract

### Background

Ethiopia has over 3,200 new cases of leprosy diagnosed every year. Prevention remains a challenge as transmission pathways are poorly understood. Susceptibility and disease manifestations are highly dependent on individual host-immune response. Nutritional deficiencies, such as protein-energy malnutrition, have been linked to reduced cell-mediated immunity, which in the case of leprosy, could lead to a higher chance of active leprosy and thus an increased reservoir of transmissible infection.

### Methodology/Principal findings

Between June and August 2018, recently diagnosed patients with leprosy and individuals without known contact with cases were enrolled as controls in North Gondar regional health centers. Participants answered survey questions on biometric data, demographics, socio-economic situation, and dietary habits. Descriptive statistics, univariate, and multivariate logisitic regression examined associations between undernutrition, specifically body mass index (BMI), middle upper arm circumference (MUAC), and leprosy. Eighty-one participants (40 cases of leprosy, 41 controls) were enrolled (75% male) with an average age of 38.6 years (SD 18.3). The majority of cases were multibacillary (MB) (90%). There was a high prevalence of undernutrition with 24 (29.6%) participants underweight (BMI <18.5) and 17 (21%) having a low MUAC. On multivariate analysis, underweight was significantly associated with leprosy (aOR = 9.25, 95% CI 2.77, 30.81). Also found to be associated with leprosy was cutting the size of meals/skipping meals (OR = 2.9, 95% CI 1.0, 8.32) or not having enough money for food (OR = 10, 95% CI 3.44 29.06).

**Data Availability Statement:** Data cannot be shared publicly because this is a small region of Ethiopia, and a small dataset of a very stigmatized

disease, therefore listing the study participants separately by age may violate participant privacy. Data are available for those who meet the criteria for access to confidential data at a third party, Professor Kassahun Alemu Gelaye, PhD, the Director, Institute of Public Health, University of Gondar, Ethiopia: kassalemu@gmail.com.

**Funding:** JF received donations from the Order of St. Lazarus (https://www.st-lazarus.us). PA and LE received funding from an institutional award: Emory RSPH Global Field Experience Award (https://www.sph.emory.edu/rollins-life/community-engaged-learning/global-field-experience/index.html). There are no grant numbers for either source. The funders had no role in study design, data collection and analysis, decision to publish, or preparation of the manuscript. No salary support was provided by these funders.

**Competing interests:** The authors have declared that no competing interests exist.

## Conclusions/Significance

The results suggest a strong association between leprosy and undernutrition, while also supporting the framework that food insecurity may lead to undernutrition that then could increase susceptibility to leprosy. In conclusion, this study highlights the need to study the interplay of undernutrition, food insecurity, and the manifestations of leprosy.

## Author summary

Understanding the effect that nutritional deficiencies, dietary habits, and undernutrition exert on leprosy transmission can improve our ability to better develop strategies and control programs to prevent this debilitating disease. While there is evidence that leprosy and undernutrition are associated, overall the literature is sparse. The authors here provide evidence for the possible role of undernutrition and low BMI on leprosy susceptibility. Additional questions about dietary habits and socioeconomic status support the framework that food insecurity may lead to undernutrition causing an increase in susceptibility to active leprosy disease. Although the study focuses on the leprosy susceptibility, as it relates to undernutrition, in North Gondar Zone, Ethiopia, the outcomes of the study may inform risks in other areas where the dual burden of undernutrition and neglected tropical diseases exist.

## Introduction

Ethiopia is endemic for leprosy (*Mycobacterium leprae* infection), with over 3,200 diagnosed every year, making it the seventh most burdened country globally [1]. Prevention remains a challenge, as transmission pathways are poorly understood and susceptibility and disease manifestations are highly dependent on the individual host immune response [2]. An immune response that predisposes an individual to clinically symptomatic leprosy, in particular multibacillary (MB) infection, could be a driver of more infectious cases in the community. Nutritional deficiencies, such as protein-energy malnutrition, have been linked to reduced cell-mediated immunity and susceptibility to infection [3–6] and food insecurity has been associated with leprosy [7–10].

Body mass index (BMI) is a measure of nutritional status in adults [11]. Another indicator for nutritional status in adults is Middle Upper Arm Circumference (MUAC) [12]. While more commonly used and validated in children, it has been used in some contexts to assess adult malnutrition [12–14]. Nutrition has long been known to influence the development of infectious diseases such as respiratory infections, infectious diarrhea, measles, and malaria [15–17]. Malnutrition often negatively affects the immune system, and in some cases, causes individuals to be more vulnerable to clinically apparent infection. In tuberculosis, another mycobacterial infection, nutritional deficit has been identified as an important risk factor for the development of clinical symptoms of disease [18–22].

Due to the fact that poverty is such a broad and complex topic to characterize, a causal relationship between poverty and leprosy is difficult to demonstrate. In addition, uncertainty exists about how leprosy and poverty are associated [8]. Most individuals affected by leprosy are born and raised in poor environments and continue to be pushed into greater depths of poverty due to the stigma and disabilities associated with the disease [9]. Low income families have little money to spend on food and consequently have a low intake of highly nutritious

non-rice foods such as meat, fish, milk, eggs, fruits and vegetables. There are few studies investigating leprosy and nutrition; however, they all suggest that nutrition plays a role in the disease [7,9,10]. We seek to increase the body of knowledge on the associations of leprosy and nutrition in the context of Ethiopia, which has not been studied thus far. We hypothesize that undernutrition is a risk factor for leprosy when controlling for *S. mansoni* co-infection (a highly prevalent helminth that also alters cell-mediated immunity) and socioeconomic status (SES).

## Methods

### Ethics statement

This study was approved by the Emory Institutional Review Board (IRB) and the ethical review board of the University of Gondar. Informed verbal consent was given by all participants.

### Study design & overview

Between June and August 2018, participants were recruited to a case-control study in North Gondar Zone, Ethiopia. North Gondar Zone is part of the Amhara Region and had a census population of 3,225,022 as of July 1, 2017 [23]. With an urban population of 509,228 (15.79%), a vast majority of the population in North Gondar Zone live in rural or agricultural areas [23]. The total land area of North Gondar Zone is 45,945 square kilometers [24]. Participants were enrolled at the University of Gondar referral hospital and health centers in and around the North Gondar Zone.

### Study population

Cases were enrolled as persons with clinical leprosy diagnosed by a practicing dermatologist within the previous 12 months. Both multibacillary (MB) and paucibacillary (PB) cases were included as per the World Health Organization (WHO): MB with greater then 5 lesions and PB less then or equal to 5 lesions [34]. Other case inclusion criteria were being 18 years of age or older and residing in North Gondar Zone of Ethiopia. Controls were adults without contact of known cases of leprosy and who resided in the same communities as the cases. Individuals with previous leprosy infection and suspicious skin or nerve symptoms were excluded as controls based on physical examination and medical history. Controls were recruited from the community, however, pregnant women and children were excluded from the study.

### Variables and measurement

Food security survey questions were derived from the United States Department of Agriculture (USDA) through the Economic Research Service (ERS) and is an accessible resource on how to measure household food security [25,26]. BMI was calculated using the definition weight (kg)/ height$^2$ (m). Underweight was defined as body mass index (BMI) less than 18.5 kg/m$^2$. Normal was classified as greater than or equal to 18.5 kg/m$^2$ and less than 25 kg/m$^2$. Overweight was classified as greater than or equal to 25 kg/m$^2$ and less than 30 kg/m$^2$ and obese as BMI greater than 30 kg/m$^2$. For MUAC, circumference was measured by finding the midpoint between the participants' elbow and shoulder. From there, the measuring tape was wrapped around the participants' arm at the midpoint to determine MUAC in centimeters. Adults with a MUAC below 18 cm were classified as "severe" or "low"risk for malnutrition. Adults with a MUAC between 18 cm and 21 cm are classified as "moderate" or "low" risk for malnutrition. Adults with MUAC greater than 21 cm are classified as "normal" risk for malnutrition [27]. Urine was tested for *Schistosoma mansoni* infection by Schisto POC-CCA rapid diagnostic test [28].

## Data collection

Data collected from participants included anthropometric data (height, weight, and middle upper arm circumberence (MUAC)), demographic information, and a questionnaire administered only in Amharic inquiring about food insecurity, dietary habits, socioeconomic status (SES), and education. Trained data collectors, health center focal persons, were responsible for collecting anthropometric parameters. Measures taken to ensure data quality include one data collector measuring height and weight at each health center, fully calibrated scales for weight measurement, and study coordinators observing accuracy of measurements.

## Sample size calculation

There are very few data to guide calculation of a sample size for the associations of low BMI (underweight, BMI < 18.5) and leprosy. Since co-infection with schistosomiasis was also a variable in this study and of interest due to the suppression of cell-mediated immunity from helminth infections, we used published data on the associations of helminths and leprosy, which showed an odds ratio of 4 for the association between helminth and leprosy, to calculate sample size for this pilot study [29]. Based on an estimated helminth burden of 20–25% prevalence in North Gondar Zone, sample size was calculated using an alpha of 0.05 and power of 0.8, resulting in a goal of 40 cases and 40 controls.

## Statistical analysis

Data from the questionnaires were entered into an Excel database. After data cleaning, analysis was performed using SAS 9.4 (Cary, NC) [30]. Descriptive statistics were performed on the main study variables and p-values describing differences between cases and controls were calculated for each variable using the chi-square, Fisher's Exact test, or t-test. In univariate analyses, calculation of odds ratios provided insight into an association between exposures of interest, such as BMI, MUAC, and dietary habits, and the outcome of interest, a clinical diagnosis of leprosy. A p-value of < 0.05 was considered significant. Figs 1 and 2 for visual analysis and representation were developed using Microsoft Excel (2017) and R [31]. After univariate analysis was carried out, the variables significantly associated with *M. leprae* infection were included in a multivariate backward stepwise logistic regression controlling for confounding factors such as age, sex, and education. The variables that remained statistically significant in these multivariate analyses were considered as the main result. All multivariate analyses were done using SAS 9.4 (Cary, NC) [30].

## Results

Eighty-one participants, 40 patients (cases) and 41 controls, were enrolled (75% male) with an average age of 38.6 years (SD 18.3). 52 participants (64.2%) had less than eight years of formal education. The majority of cases were MB (90%) and 21 participants (25.9%) had *S. mansoni* infection. All demographic and clinical data are presented in Table 1.

Eight (20% of cases) individuals were diagnosed with Grade II disability, according to WHO classification. Disability Grade II was highest among patients between the age of 39–59 accounting for 36% of cases within that group. BMI ranged from 15.9 to 31.5 with a high prevalence of undernutrition: 24 (29.6%) participants underweight (BMI <18.5) and 17 (21%) with "low" MUAC. Four (9.8%) controls were characterized as underweight, while 20 (50%) cases were classified as underweight by BMI. Fig 1 depicts a comparison of BMI among cases and controls showing that the cases' mean BMI is statistically significantly lower than that of

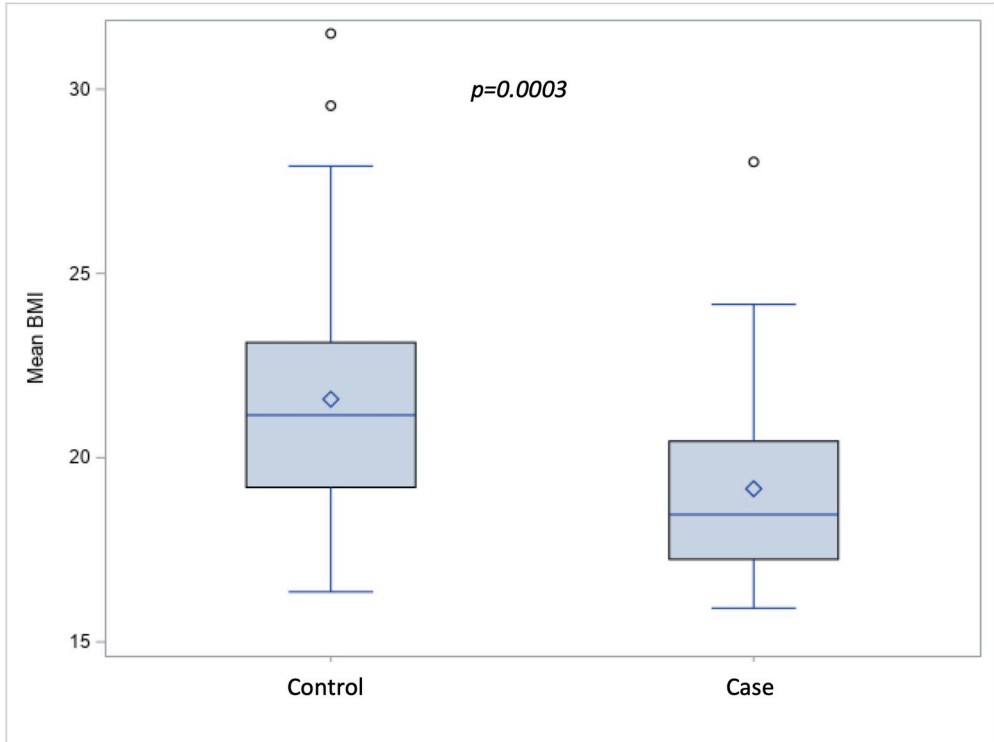

**Fig 1. Comparison of mean BMI (kg / m$^2$) between cases and control.**

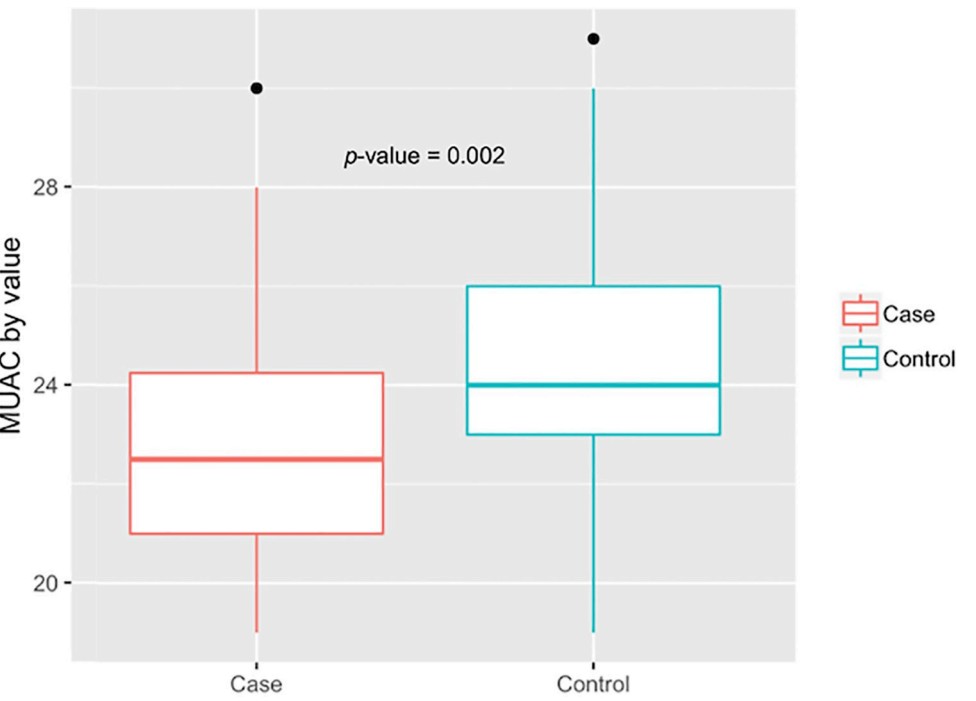

**Fig 2. Comparison of mean MUAC (cm) between cases and controls.**

**Table 1. Demographic and clinical data of study population.** P-values determined by chi-square, t-test, or crude logistic regression where appropriate. An alpha = 0.05 was considered statistically significant.

| Variable | Cases (n = 40) | Controls (n = 41) | Total (n = 81) | p-value |
|---|---|---|---|---|
| Age (years), mean (SD) | 42.9 (19.1) | 34.4 (16.5) | 38.6 (18.3) | p = 0.04 |
| Sex, n (%) | | | | p = 0.66 |
| Male | 31 (77.5) | 30 (73.2) | 61 (75.3) | |
| Female | 9 (22.5) | 11 (26.8) | 20 (24.7) | |
| WHO Classification, n (%) | | N/A | N/A | |
| Paucibacillary (PB) | 4 (10.0) | | | |
| Multibacillary (MB) | 36 (90.0) | | | |
| Grade of Disability, n (%) | | N/A | N/A | |
| Grade 1 | 12 (30.0) | | | |
| Grade 2 | 8 (20.0) | | | |
| Education Level, n (%) | | | | p = 0.02 |
| Less than Grade 8 | 27 (67.5) | 25 (61.0) | 52 (64.2) | |
| Grade 8 and above | 4 (10.0) | 15 (36.6) | 19 (23.5) | |
| Missing | 9 (22.5) | 1 (2.4) | 10 (12.3) | |
| Body Mass Index, n (%) | | | | p = 0.0004 |
| Underweight (<18.5) | 20 (50.0) | 4 (9.8) | 24 (29.6) | |
| Normal (18.5–24.9) | 19 (47.5) | 34 (82.9) | 53 (65.4) | 1 (ref) |
| Overweight–Obese (≥25) | 1 (2.5) | 3 (7.3) | 4 (4.9) | p = 0.66 |
| Middle Upper Arm Circumference, n (%) | | | | P = 0.003 |
| Low (18–21 cm) | 14 (35.0) | 3 (7.3) | 17 (21.0) | |
| Normal (>21 cm) | 26 (65.0) | 38 (92.7) | 64 (79.0) | |
| S. mansoni infection, n (%) | 8 (20.0) | 13 (31.7) | 21 (25.9) | P = 0.12 |

controls' mean BMI. Fig 2 depicts a comparison of MUAC among cases and controls, with cases having lower mean MUAC.

On univariate analysis, both low BMI (OR = 9.25, 95% CI 2.77, 30.81) and low MUAC were significantly associated with leprosy (OR = 6.82, 95% CI 1.78, 26.13). Low education level, defined as less than Grade 8, was also significantly associated with leprosy (OR = 4.05, 95% CI 1.18, 13.85). Cutting the size of meals/skipping meals (OR = 2.87, 95% CI 1.0, 8.32) or not having enough money to get more food (OR = 10, 95% CI 3.44, 29.06) was more common in cases of leprosy than controls. Additional outcomes looking at dietary habits and SES as they relate to leprosy can be seen in Table 2. On multivariate analysis (Table 2), underweight was still significantly associated with leprosy (aOR = 10.32, 95% CI 1.79, 59.67) after controlling for age, sex, and education (as a surrogate marker of socioeconomic status). MUAC, also included in the multivariate model, was no longer statistically significantly associated with leprosy (aOR = 1.50, CI 0.22, 10.05).

## Discussion

Undernutrition, as defined by low body mass index, showed a striking association with leprosy in this small case control study with an aOR of more than 10. Wagenaar, et al. and Oktaria et al, also both found BMI to be associated with leprosy [7]. More specifically, Wagenaar, et al., found 25% of cases to be underweight, while only 14% of controls were found to be underweight [7]. According to Oktaria, et al., a paired t-test showed a significant difference in BMI between cases and controls [10]. Although both studies' findings were consistent with our findings overall, their calculations used mean BMI as a continuous variable to compare between cases and controls. By grouping underweight (BMI < 18.5) to Normal–Obese (BMI ≥ 18.5), we have a better picture of the degree of undernutrition in many of the participants, especially those with leprosy. These findings are consistent with Rao, et al., who showed

**Table 2. Univariate analysis and multivariate logistic regression of study outcome, leprosy, and demographic, nutritional, infectious, and food security questions.** Variables without an aOR were either not included or were dropped from the final model per the methods.

| Variable | OR (95% CI) | P-Value | aOR (95% CI) |
|---|---|---|---|
| Underweight–BMI (<18.5) | 9.25 (2.77, 30.81)* | <0.0001 | 10.32 (1.79, 59.67)* |
| Low–MUAC (≤ 21 cm) | 6.82 (1.78, 26.13)* | 0.003 | 1.50 (0.22, 10.05) |
| Sex<br>    Male (ref = Female) | 1.26 (0.46, 3.48) | 0.66 | 2.35 (0.51, 10.87) |
| Age, years, continuous | ---- | ---- | 1.01 (0.98, 1.05) |
| Education level (Less than Grade 8) | 4.05 (1.18, 13.85)* | 0.02 | 1.87 (0.41, 8.52) |
| No utilization of institutional banking | 2.89 (1.17, 7.14)* | 0.02 | |
| *S. mansoni* infection | 0.54 (0.19, 1.49) | 0.12 | |
| Reducing or skipping meals | 2.87 (1.00, 8.32)* | 0.05 | |
| Insufficient funds for meals | 10.0 (3.44, 29.06)* | < 0.001 | |
| Length of time between each market visit?<br>    Less than once a week<br>    (ref = at least once a week) | 1.83 (0.71, 4.71) | 0.22 | |
| Did not eat for a day due to lack of food | 1.86 (0.50, 6.82) | 0.37 | |
| Ate less than participant felt they should have | 1.27 (0.42, 3.84) | 0.68 | |

*Result is significant with a p-value <0.05.

that undernutrition (BMI < 18.5) was more common in people affected by leprosy than in those without leprosy in India [32]. A commonality among all the studies is a high burden of undernutrition, as defined by low BMI, that was present within "case" study populations. Since all these studies are taken at one point in time, it is not possible to know whether the undernutrition predated the infection or not. However, since poor micronutrition and several micronutrient deficiencies, such as vitamin A, can suppress cell-mediated immunity, these findings could show a propensity for undernourished individuals to either present with active leprosy symptoms (as opposed to latent infection) or to present with multibacillary infection instead of paucibacillary infection. In both cases, undernutrition would be sustaining a reservoir of infection in the community. Feenstra et al.'s study supports the notion of malnutrition being a risk factor for leprosy as they found a higher incidence following periods of food shortage and more longitudinal studies need to investigate this further [9].

Another important outcome of this study revealed that food insecurity, measured by having insufficient funds for meals, was also associated with leprosy on univariate analysis. While, again, it is hard to know if the leprosy itself caused the food insecurity (perhaps through stigma or social isolation), there may be a case that having food insecurity leads to malnutrition that then increases the risk of active leprosy. This was consistent with Wagenaar, et al., who found that low income families have only little money to spend on food and consequently have a low intake of highly nutritious non-rice foods such as meat, fish, milk, eggs, fruits and vegetables and thus at higher risk for leprosy infection [7].

The majority of cases being MB (90%) within the study population may highlight comorbid conditions such as undernutrition that may play a role in the shift from PB to MB among patients, since MB is associated with a lack of a sufficient cell-mediated (Th1) response as opposed to a robust cell-mediated response in PB cases [33]. Although numbers of PB cases were too low to study this question directly, further investigation into the association with malnutrition and presentation of leprosy (MB vs PB) is warranted since such much higher rates of transmission occur from patients with MB than PB leprosy [34]. In addition to the MB proportion of patients, about 21 participants (25.9%) in this study presented with *S. mansoni*

infection. Although our study did not find an association with *S. mansoni* infection and leprosy, further investigations into co-infections and the influence on immune response mechanisms is needed to better understand this interaction with leprosy as several studies have shown a connection, including a recent one in Brazil that showed a significant association between leprosy and schistosomiasis between cases and household / family contacts [35].

Findings from this study support the limited published data on nutrition and leprosy and presents the first study to look at nutrition and leprosy in the Ethiopian context, a country that has dealt with a lot of food insecurity for its citizens in recent decades. A major strength of this study is the survey combining anthropometric data and nutritional questions related to dietary habits and food insecurity. One limitation of the study is the smaller study size. However, with regard to BMI, sample size was sufficient to validate our findings with 80% power at a 0.05 alpha level. Challenges with sample size stemmed from a limited number of reported cases, accessibility to health centers, and willingness of patients to enroll in the study. Additionally, this study did not measure micronutrient deficiencies in participants. Doing so may have provided a more comprehensive study that would add to the literature. Although *S. mansoni* co-infection data was collected, other co-infections were not studied that could potentially affect immune responses, and thus the risk of active leprosy. While this study found an association with BMI and leprosy, further knowledge and understanding is needed to corroborate the idea that there is a causal pathway between undernutrition and leprosy. As Wagenaar also argues the fact that food shortage may not impact susceptibility to leprosy infection itself irrespective of *M. leprae* manifestation, but rather the progression from latent infection to clinical manifestation is important to consider [7]. In other words, a better understanding of the individual host response to latent *M. leprae* and contribution of comorbidities to this response is crucial to controlling leprosy and reducing the human reservoirs of infection in affected areas.

This study is the first of its kind to look at nutrition, food insecurity, and SES in the context of North Gondar Zone, Ethiopia. Addressing the interplay between poverty, undernutrition, dietary habits, and nutritional deficiencies (as shown in Fig 3) may have the potential for impact on leprosy burden in this region and similar endemic areas. As we begin looking

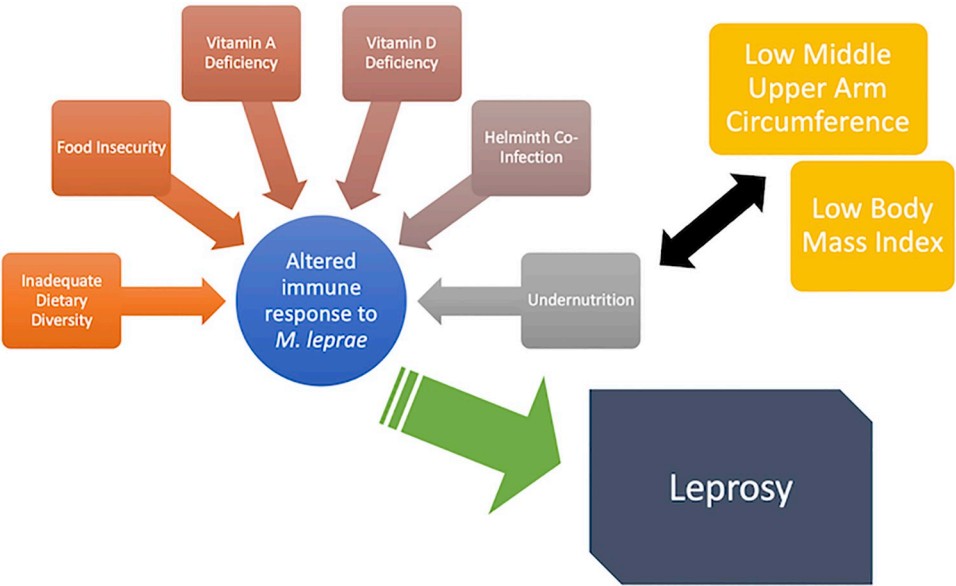

**Fig 3. Diagram showing possible relationships between food insecurity, dietary diversity, nutritional deficiencies, undernutrition, *M. leprae* infection, and leprosy manifestation.**

forward toward the elimination of leprosy, there are two major components that must be addressed; the reservoir (humans, zoonotic, and possibly environmental) and the host. This study has focused primarily on the host, in terms of undernutrition and dietary habits' influence on leprosy susceptibility, infection, and/or development of leprosy in a specific population. It is a worthwhile endeavor to pursue a greater understanding of the interconnectedness of nutrition and leprosy to further improve leprosy control, prevention, and intervention.

## Acknowledgments

The authors thank Yawyewsew Alemu for assistance in coordinating health center visits and Workye Kassaw for printing and translating services. We also appreciate the assistance of health center staff and thank the participants without which the study could not have been done.

## Author Contributions

**Conceptualization:** Puneet Anantharam, Kassahun D. Bilcha, Jessica K. Fairley, Annisa B. Tesfaye.

**Data curation:** Puneet Anantharam, Lisa E. Emerson, Annisa B. Tesfaye.

**Formal analysis:** Puneet Anantharam, Jessica K. Fairley.

**Funding acquisition:** Puneet Anantharam, Jessica K. Fairley.

**Investigation:** Puneet Anantharam, Lisa E. Emerson, Jessica K. Fairley, Annisa B. Tesfaye.

**Methodology:** Puneet Anantharam, Jessica K. Fairley, Annisa B. Tesfaye.

**Project administration:** Kassahun D. Bilcha, Annisa B. Tesfaye.

**Resources:** Kassahun D. Bilcha, Annisa B. Tesfaye.

**Supervision:** Kassahun D. Bilcha, Jessica K. Fairley, Annisa B. Tesfaye.

**Writing – original draft:** Puneet Anantharam.

**Writing – review & editing:** Lisa E. Emerson, Kassahun D. Bilcha, Jessica K. Fairley, Annisa B. Tesfaye.

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
