## [Decision Letter · Decision Letter 0]

29 Dec 2020

Dear Dr Fairley,

Thank you very much for submitting your manuscript "Undernutrition, food insecurity, and leprosy in North Gondar Zone, Ethiopia: A case-control study to identify infection risk factors associated with poverty" for consideration at PLOS Neglected Tropical Diseases. As with all papers reviewed by the journal, your manuscript was reviewed by members of the editorial board and by several independent reviewers. In light of the reviews (below this email), we would like to invite the resubmission of a significantly-revised version that takes into account the reviewers' comments below. 

We cannot make any decision about publication until we have seen the revised manuscript and your point by point response to the reviewers' comments. Your revised manuscript is also likely to be sent to reviewers for further evaluation.

Sincerely,

Zulfiqar A. Bhutta, PhD

Associate Editor

Abhay Satoskar

Deputy Editor

Reviewer's Responses to Questions

**Key Review Criteria Required for Acceptance?**

**Methods**

-Are the objectives of the study clearly articulated with a clear testable hypothesis stated?

-Is the study design appropriate to address the stated objectives?

-Is the population clearly described and appropriate for the hypothesis being tested?

-Is the sample size sufficient to ensure adequate power to address the hypothesis being tested?

-Were correct statistical analysis used to support conclusions?

-Are there concerns about ethical or regulatory requirements being met?

Reviewer #1: Methods

1. It is unclear how the authors ascertain controls. On page 6, 108 -109 It is stated “Controls were adults without contact of known cases of leprosy and who resided in the same communities as the cases…..Individuals with previous leprosy infection and suspicious skin or nerve symptoms were excluded as controls” Is this enough to ascertain controls? This sentence could further be expanded to include if any physical examination and/or any laboratory confirmations were conducted. This is important because it could have led to misclassification. In addition, it is important to describe how controls were recruited i.e. from facility or community?

2. Measurement of variables is not clearly stated. It is explained, to some extent, how MUAC was measured. However, measurement procedures for other anthropometric parameters i.e. height and weight should be explained in detail including if there is training of data collectors, measures taken to ensure data quality, technical error measurements during training (if calculated) and other procedures followed while measuring height and weight need to be clearly explained. Using a separate paragraph for each measurement (anthropometric, dietary and others) under the headline of variables/measurements would improve organization. 

3. I found it difficult to link the results of the study with the USDA food security measurement tool the authors described in the data collection section. The collected food security variables doesn’t seem to appropriately measure food security. Moreover, the authors might want to explain why they chose this specific tool to assess food security. On quick literature review, there are evidences of a valid household food insecurity access scale to assess food insecurity including seasonality in Ethiopia (such as Gebreyesus et al. BMC Nutrition 2015, 1:2 http://www.biomedcentral.com/bmcnutr/content/1/1/2 and Kabalo et al. BMC Nutrition (2019) 5:54 https://doi.org/10.1186/s40795-019-0323-6). 

4. The sources of measurements (data collection tool) used to assess dietary habit is also not clear and it is very difficult to say the variables stated in table 2 measure dietary habit. Table 2; what does “recently modified dietary intake mean”? Is this increasing or decreasing dietary intake? By how much? How it was measured need to be clearly stated. Is it common to take dietary supplements in Ethiopia, particularly in the study setting?

5. The data analysis section mainly talks about sample size. It would be better understandable if sample size and data analysis were described as separate sections. 

6. On page 7, line 147-148; It is stated “education (found to be the most useful marker of socioeconomic status in our study).” How did the authors reach to this conclusion? Also page 9 line 181 stated “education (as a surrogate marker of socioeconomic status)“. These statements seem debatable considering it is stated, in the study setting section, that the vast majority of the population in the study area live in agricultural areas. 

7. Why is it that socioeconomic variables i.e. wealth index or at least monthly family income were not measured?

Reviewer #2: Given that this is a case control study, the selection of control is critical. Need more details of how the controls were selected. Currently a superficial description is given that "Controls were adults without contact of known cases of leprosy and who resided in the same communities as the cases". Were they randomly selected from the community? Were they enrolled from the clinics with alternate diagnosis? 

Page 6 L125. Low risk should actually be high risk.

Reviewer #3: Major revision

Helminth infection is described in the introduction as having an effect on the cell mediated immunity, what according to theory is also the case of undernutrition in relation to leprosy. Therefore, Helminth infection should be taking into account as a confounding factor in the analysis (in all univariate analyses, and the multivariate analyses). 

The hypothesis at the end of the introduction suggests that the analyses are controlled for co-infection, but from the rest of the text it does not become clear if this was indeed done. If it is not done, it should be. If it was done, it should be better described in the methods section.

**Results**

-Does the analysis presented match the analysis plan?

-Are the results clearly and completely presented?

-Are the figures (Tables, Images) of sufficient quality for clarity?

Reviewer #1: Result

1. It looks like majority (75%) of the participants were males. This number is almost similar between cases and controls. Is this a coincidence or did the authors match cases and controls by sex? In case of the later, the authors need to consider appropriate analysis.

2. Results of a characteristics comparison between cases and controls is not included for each variable as stated in the analysis although there is a figure for MUAC and BMI. It would be easy to understand if p-vaues were written next to each variable embedded in table 1. 

3. While classifying BMI, Normal and Obese individuals should not be in the same category. 

4. Presenting the univariate and multivariate analysis results in one table, by clearly indicating the reference categories for each of the variables would improve understanding.

Reviewer #2: Figure 1 and 2 are exactly the same as Table 3. Should use one of the two.

Reviewer #3: Accept

Figure 1 and 2 do not add much to the information already given in Table 1. I suggest to leave those out, as they are also not of good quality..

**Conclusions**

-Are the conclusions supported by the data presented?

-Are the limitations of analysis clearly described?

-Do the authors discuss how these data can be helpful to advance our understanding of the topic under study?

-Is public health relevance addressed?

Reviewer #1: (No Response)

Reviewer #2: We need better understanding of control selection to say how valid are the results.

Reviewer #3: Accept

It would be interesting to discuss the results of the food insecurity questionnaire a bit further. What could be an explanation that there is a strong correlation between insufficient funds for foods with leprosy, but not with the other food related factors (reduced meals or skipping meals for a day- which you expect to be strongly related to insufficient funds, and eating less than should have).

**Editorial and Data Presentation Modifications?**

Reviewer #1: Some minor comments

1. Expand abbreviation in the abstract 

2. Page 6, line 108; “Other case inclusion criteria were being 18 years of age or older or residing in North Gondar Zone of Ethiopia” is it if a person fulfills either one of the 2 criteria or both?

3. Page 6, line 114; was the demographic questions asked in a language other than Amharic? 

4. Page 6, 119-21; write measurement unit next to the numbers 

5. The sentence “A p-value of < 0.05 was considered significant.” Is repeated on page 7 lines 143 and 150. I think it is enough to just keep one. Same goes for the analysis software. 

6. Page 8, lines 63-65; the numbers in table 2 and sentences doesn’t match.

Reviewer #2: (No Response)

Reviewer #3: Minor revisions:

There are a few phrases that could be better formulated in English:

Line 32: " recently diagnosed patients and individuals without contact of known cases were enrolled as controls" suggests that patients were also enrolled as controls. Suggest to rephrase as and "recently diagnosed leprosy patients and controls, i.e. individuals ...., were enrolled."

Line 34: ": Biometric data" should be "anthropometric data"

Line 104: "Cases were enrolled as persons with active leprosy disease" is not correct. Suggest to rephrase as "persons with active leprosy disease were enrolled as cases".

Line 108: "years of age or older or residing" should be "and residing"

Line 108/109: "Controls were adults without contact of known cases of leprosy". Rephrase to either "Controls were adults who were not a contact of leprosy cases" or "Controls were adults without contact with known leprosy cases"

Line 125 and 126: "or “low”risk for malnutrition" is very unclear. With such low MUAC scores it seems not likely that the risk for malnutrition is low.

Line 182 ends unexpectedly

**Summary and General Comments**

Reviewer #1: Overall comment

I’m grateful for the opportunity to review your manuscript entitled “Undernutrition, food insecurity, and leprosy in North Gondar Zone, Ethiopia: A case-control study to identify infection risk factors associated with poverty”. The study addressed an interesting issue and could be informative especially with little evidence available on the issue in Ethiopia. The submitted manuscript unfortunately requires major revision before it is ready for publication, as it lacks methodological detail that would facilitate sound interpretation of the results and permit reproducibility of the study elsewhere (details below).

Reviewer #2: Important topic, simple design. Need to strengthen the methods and results section.

Reviewer #3: This is a clearly written paper on the relationship between undernutrition and leprosy. The paper is a relevant addition to the knowledge so far, as it studies a different population than earlier papers.

PLOS authors have the option to publish the peer review history of their article (what does this mean?). If published, this will include your full peer review and any attached files.

Reviewer #1: No

Reviewer #2: Yes: Syed Asad Ali

Reviewer #3: No
---

## [Editor Report · Decision Letter 1]

8 May 2021

Dear Dr Fairley,

We are pleased to inform you that your manuscript 'Undernutrition, food insecurity, and leprosy in North Gondar Zone, Ethiopia: A case-control study to identify infection risk factors associated with poverty' has been provisionally accepted for publication in PLOS Neglected Tropical Diseases.

Best regards,

Zulfiqar A. Bhutta, PhD

Associate Editor

Abhay Satoskar

Deputy Editor

---

## [Editor Report · Acceptance letter]

21 Jun 2021

Dear Dr Fairley,

We are delighted to inform you that your manuscript, "Undernutrition, food insecurity, and leprosy in North Gondar Zone, Ethiopia: A case-control study to identify infection risk factors associated with poverty," has been formally accepted for publication in PLOS Neglected Tropical Diseases.

Best regards,

Shaden Kamhawi

co-Editor-in-Chief

Paul Brindley

co-Editor-in-Chief
